# Creating an AI-Enhanced Morse Code Translation System Based on Images for People with Severe Disabilities

**DOI:** 10.3390/bioengineering10111281

**Published:** 2023-11-03

**Authors:** Chung-Min Wu, Yeou-Jiunn Chen, Shih-Chung Chen, Sheng-Feng Zheng

**Affiliations:** 1Department of Intelligent Automation Engineering, National Chin-Yi University of Technology, Taichung 411030, Taiwan; cmwu@ncut.edu.tw; 2Department of Electrical Engineering, Southern Taiwan University of Science and Technology, Tainan 710301, Taiwan; chenyj@stust.edu.tw; 3Department of Intelligent Robotics Engineering, Kun Shan University, Tainan 710303, Taiwan; s110001321@g.ksu.edu.tw

**Keywords:** artificial intelligence, communication, Morse code, disabilities, caregiver

## Abstract

(1) Background: Patients with severe physical impairments (spinal cord injury, cerebral palsy, amyotrophic lateral sclerosis) often have limited mobility due to physical limitations, and may even be bedridden all day long, losing the ability to take care of themselves. In more severe cases, the ability to speak may even be lost, making even basic communication very difficult. (2) Methods: This research will design a set of image-assistive communication equipment based on artificial intelligence to solve communication problems of daily needs. Using artificial intelligence for facial positioning, and facial-motion-recognition-generated Morse code, and then translating it into readable characters or commands, it allows users to control computer software by themselves and communicate through wireless networks or a Bluetooth protocol to control environment peripherals. (3) Results: In this study, 23 human-typed data sets were subjected to recognition using fuzzy algorithms. The average recognition rates for expert-generated data and data input by individuals with disabilities were 99.83% and 98.6%, respectively. (4) Conclusions: Through this system, users can express their thoughts and needs through their facial movements, thereby improving their quality of life and having an independent living space. Moreover, the system can be used without touching external switches, greatly improving convenience and safety.

## 1. Introduction

According to the World Report on Disability [1] issued by the World Health Organization, the proportion of the global population aged 60 and over is projected to double from 11% to 22%, with an estimated 15% of the world’s population experiencing disabilities by 2050. Severe disability encompasses conditions like quadriplegia, wherein individuals with significant health issues or disabilities, such as impairments in movement and speech, are confined to a bed throughout the year. They are unable to perform most actions, limited to only basic functions like blinking, moving their cheeks, or twitching their fingers. Their daily existence primarily involves gazing at the ceiling and contemplating. Individuals facing severe disabilities lack the capacity to move independently or communicate with others. However, their sensory and autonomic nervous systems remain unaffected, preserving their hearing, vision, touch, smell, taste, thinking, and cognitive abilities. Care robots (CRs) are transforming the processes of care, therapy, assistance, and rehabilitation. The Policy Department for Economic, Scientific and Quality of Life Policies of the European Parliament categorized the CRs into the following four groups: robotic surgery, care and socially assistive robots, rehabilitation systems, and training for health and care workers [2].

This study is part of the field of assistive robots for care and social interaction, where connecting with others or operating machines is a huge challenge for those with severe disabilities due to their limitations. “Communication” denotes the process of sharing thoughts, transmitting information, and establishing connections. Effective communication relies on appropriate communication tools or media. These tools encompass language, written text, bodily movements, facial expressions, and various interactive information formats. Certain profound disabilities, such as amyotrophic lateral sclerosis (ALS), motor neuron disease (MND), cerebral palsy (CP), spinal cord injury (SCI), and post-stroke intubation, can lead to enduring impairments in both speech and mobility. These conditions can present significant barriers to engaging with the external world, leading to challenges in articulation, the inability to communicate directly, and even the potential for misunderstandings to arise between caregivers and individuals living with these disabilities. Individuals like those afflicted with ALS, MND, CP, SCI, and post-stroke intubation can significantly enhance their quality of life through the assistance of augmentative communication systems [3]. Therefore, the development of assistive communication systems holds paramount importance in enhancing communication capabilities of individuals with disabilities. In summary, resolving the communication predicament stands as the most pressing issue, promising to infuse greater meaning and happiness into the lives of those severely disabled.

In recent times, computers have become an integral part of contemporary life, serving essential roles in work, entertainment, and daily routines. Nevertheless, for individuals afflicted with severe physical disabilities, operating a computer often presents a formidable challenge. Thanks to the rapid progression of technology, numerous augmentative and alternative communication (AAC) solutions have emerged as viable options to facilitate communication and interaction for these patients [3,4,5,6,7,8,9,10,11,12,13,14,15,16,17,18,19,20,21,22,23,24,25,26,27,28,29]. These innovative alternatives encompass a wide array of approaches, including eye tracking [8,9], head control devices [10,11,12], infrared control devices [12,13], voice control systems [14], auxiliary physiological signal devices like electrooculogram (EOG) switches [15,16,17], electromyography (EMG) switches [18,19,20], electroencephalography (EEG) devices [21,22,23,24,25,26,27], and scanning auxiliary input tools [28]. Eye-tracking technology has shown substantial contributions in various research areas particularly in health care, education, and industrially. Eye tracking has been able to provide valuable support for individuals with severe disabilities as a useful tool for human–computer interaction [9]. A head-operated computer mouse employs two tilt sensors placed in the headset to determine the head position and function as a simple head-operated computer mouse. One tilt sensor detects the lateral head motion to drive the left/right displacement of the mouse. The other one detects the head’s vertical motion to move up and down with respect to the displacement of the mouse. A touch switch device was designed to contact gently with the operator’s cheek. The operator may puff his cheek to trigger the device to perform single click, double clicks, and drag commands [10]. An eyeglass-type infrared (IR)-controlled computer interface for the disabled may serve to assist those who suffer from spinal cord injuries or other handicaps in operating a computer. This design use of an infrared remote module fastened to the eyeglasses could allow the convenient control of the input motion on the keys of a computer keyboard and mouse which are all modified with infrared receiving/signal-processing modules [13]. Physiological signals such as EOG, EMG, and EEG can be instrumental in solving communication or computer control problems for severely disabled patients. However, these signals are susceptible to interference from environmental factors, leading to system instability. Scanning auxiliary input tools are polling-based devices, exemplified by the Assistive Context-Aware Toolkit developed by Intel for Dr. Hawking, where users only need to use controllable parts of their body to operate it. One drawback is its relatively slower operation [28]. Additionally, some AAC options incorporate communication boards integrated into AAC devices, enabling users with speech impairments to communicate by interacting with predefined commands [4]. Moreover, AAC applications have been developed to empower users to express their thoughts, needs, and ideas through speech notation technology [5]. The introduction of the tongue drive system has significantly improved the interaction between individuals with severe disabilities and their surroundings [6]. In parallel, certain systems rely on the recognition of specific breathing patterns to convey pre-established words, simplifying the process of patients expressing their needs [7]. Dr. Wu, the author of this study, has also played a pioneering role in the development of the Morse code Translator (MCT), aiming to address accessibility issues for individuals with severe disabilities [29]. Regrettably, some of the AAC devices mentioned earlier [6,7,8,9,10,11,12,13,14,15,16,17,18,19,20,21,22,23,24,25,26,27,28,29] necessitate users to wear them and rely on caregivers for their operation. In the absence of assistance, patients are left with no alternative but to wait.

Over the past decade, due to the advancement of machine learning technology, the reduction in computer computation and storage costs, and the accumulation of extensive data sets, the field of artificial intelligence has undergone rapid development. This growth has resulted in numerous breakthroughs in practical applications, including automated financial transactions, real-time financial anomaly detection, and the widespread adoption of self-driving vehicles and drones. These achievements underscore the vast potential of artificial intelligence technology. In an effort to address the issue of severely disabled patients relying on caregivers to assist them in donning and configuring AAC equipment, this research project aims to create a contactless assistive communication system using artificial intelligence image recognition modules. This groundbreaking system enables users to interact with external devices without the need for physical contact. It employs facial motion recognition to replace traditional keyboard and mouse functions, granting users independent control over computers and various 3C products to meet their daily needs.

## 2. Materials and Methods

This study introduces a Morse code translation system (AIMcT) based on artificial intelligence enhanced images. The system uses artificial intelligence technology to extract facial features [30,31], encode facial movements, and then convert them into keyboard or mouse control commands through fuzzy time recognition algorithms to achieve computer interaction. The design approach to the AIMcT system is outlined below, and Figure 1 illustrates the architecture of an AIMcT system designed for severe disabilities.

### 2.1. Face Feature Detection

The system uses the webcam to obtain the user’s face image, OpenCV for basic image processing, the face anchor provided by Dlib to obtain the user’s detailed face features, and finally sets the specific action to trigger the input signal according to Morse code.

#### 2.1.1. Image Straightening

When the image captured with the camera is tilted, the system will automatically adjust the image to the vertical direction so that Dlib’s image feature calculation can operate normally. When using Dlib to obtain the subject’s facial feature anchor point, the confidence value returned by the Dlib function is used as the basis for whether to rotate the image. The confidence value is a number between 0 and 1. The system uses the function provided by OpenCV to rotate the image until the system obtains a confidence value of the face anchor point higher than the threshold, and then stops rotating the image. This system sets the image rotation correction threshold to 0.7, and the angle of each image rotation is ±5 degrees.

#### 2.1.2. Image Compensation

In the image capture part of this study, a webcam with only 2 million pixels or more is required. The effectiveness of the AIMcT system might be susceptible to factors like the surrounding lighting conditions. This variable could affect the overall accuracy of the AIMcT system’s outcomes. Regarding the impact of lighting, OPENCV provides relevant compensation functions designed to mitigate the destabilizing effects of varying light conditions on the system’s reliability. In this study, logarithmic transformation and gamma transformation were employed for compensation in situations of excessive darkness and excessive brightness, respectively.

Logarithmic transformation serves to expand the low gray value portion of the image, thereby revealing more details in that range, while compressing the high gray value portion to reduce excessive detail. This approach emphasizes the low gray value portion of the image, as described in Equation (1).
S = c × log_v + 1_(1 + v × r)   r ∈ [0,1](1)

On the other hand, gamma transformation primarily serves for image correction, rectifying images with excessively high or low gray values to enhance contrast, as expressed in Equation (2).
S = c × rγ      r ∈ [0,1](2)

S is the output gray level of the pixel, c and γ are constants, r is the input gray level of the pixel, and v + 1 is the base number.

This adjustment is aimed at enabling the Dlib module to acquire optimal facial recognition images, thereby enhancing the system’s ability to accurately detect and track facial movements.

#### 2.1.3. Dlib Module

Dlib is a modern C++ toolkit that encompasses machine learning algorithms and tools designed for developing complex software in C++ to address real-world problems. In the AI Face feature extraction component, we utilized the Dlib image recognition module (version 19.8.1) [30], as depicted in Figure 2, to identify the mouth and eye regions as areas for autonomous movements. A facial movement recognition algorithm was created using 68 feature points, with P0–P16 representing the facial contours, P17–P26 denoting the eyebrows, P27–P35 corresponding to the nose, and P36–P47 and P42–P47 assigned to the right and left eyes, respectively. The mouth region is delineated by P48–P54, which represent the upper lip contours, P61–P63 for the inner contours of the upper lip, P55–P60 and P64 for the outer contours of the lower lip, and P65–P67 capturing the inner contours of the lower lip.

While the mouth was in motion, we calculated the distance h between P62 and P66 (Equation (3)). When h exceeds 10 pixels, it indicates an open mouth; otherwise, it signifies a closed mouth. We recorded the duration of both mouth opening and closing. We utilized the duration of both opening and closing times to dynamically adjust the judgment threshold for distinguishing between long and short durations through a fuzzy algorithm, thereby facilitating automatic identification. For instance, if the opening time falls below the threshold, it is categorized as a short opening time (dot), while durations above the threshold are classified as long opening times (dash). Subsequently, movement encoding is based on the amalgamation of long and short opening times, where a brief closing time signifies a command combination, while a prolonged closing time signifies a command output.
(3)$$h=\sqrt{{\left(P_{66x}-P_{62x}\right)}^{2}+{\left(P_{66y}-P_{62y}\right)}^{2}}$$

#### 2.1.4. Fuzzy Time Recognition Algorithm

The artificial intelligence-enhanced Morse code translation system’s management of image command combinations hinges on the identification of mouth opening and closing states, as well as the duration of the intervals between them. The artificial intelligence Dlib module is used to detect the mouth opening and closing status. To enhance input efficiency and accuracy, and maintain the stability of these command combinations, the system dynamically adjusts the time threshold for opening/closing mouth movements. This study employed a fuzzy time recognition algorithm (FTR) for precise time threshold fine-tuning. Figure 3 illustrates the differentiation between long (dash) and short (dot) signals, determined by the duration between mouth opening and closing. The system combines the durations of closing and opening times to generate command combinations. For instance, if a long opening time repeats three times, the system produces the letter “o.” This system can generate characters as specified in the Morse code table [27].

The FTR [29] is described as follows:

The block diagram of the fuzzy motion recognition algorithm is shown in Figure 4. The variable z^−1^ is a unit delay for the next step.

For the purpose of achieving stable and effortless typing, the duration between mouth opening and closing, designated as the input signal *I(k)*, undergoes normalization and constraint through the limitation function (*LF*).
(4)$$LF=\left\{\begin{array}{c} x\left(k\right)=I\left(k\right),\;I(k)<{threshold}_{k}\\ x\left(k\right)=\frac{1}{3}I(k),\;I(k)\geq {threshold}_{k} \end{array}\right.$$

For the kth frame, *X*(*k*), the prediction error e_k_ of a fuzzy algorithm is estimated as
(5)$$e_{k}=\frac{\;X(k)-Y(k-1)}{L},$$
where *Y*(*k* − 1) is the threshold at the *k* − 1 frame. The initial threshold value, *Y*(0), is predefined. It can be estimated using the average power energy of the first M-frames. *L* is the error tolerance range, which we define as 100 ms, when e_k_ is larger than 1 or smaller than −1. Thus e_k_ is 1 or −1, respectively.

This study employed linguistic rules to establish the relationship between input and output. In the process of fuzzification and defuzzification, fuzzy sets A and B were utilized, with the input range of the fuzzifier and the output range of the defuzzifier spanning from −1 to 1. To ensure algorithm performance and stability, a total of five fuzzy sets were used. These five fuzzy sets are defined as follows: negative large (LN), negative small (SN), zero (ZE), positive small (SP), and positive large (LP). The fuzzifier operated across the range from A_1_ to A_5_, where A_1_ represents LN, A_2_ stands for SN, A_3_ corresponds to ZE, A_4_ signifies SP, and A_5_ denotes LP. Similarly, the defuzzifier covered the range from B_1_ to B_5_, with B_1_ representing LN, B_2_ for SN, B_3_ for ZE, B_4_ for SP, and B_5_ for LP. According to the fuzzy set calculations, the fuzzy inference rules are as follows.
(6)$$\mathrm{If}\;e_{i}\;\mathrm{is}\;A_{i}\;\mathrm{then}\;e_{k}^{\prime }\;\mathrm{is}\;B_{i},\;i=1,2,\ldots ,5,$$
where e_i_ is the input variable of fuzzifier in the fuzzy sets A and $e_{k}^{\prime }$ is the output variable of the defuzzifier in the fuzzy sets B.

Following this, a defuzzification process was employed to yield a finite output number. In this study, the center of gravity method was utilized to calculate the output variable, $e_{k}^{\prime }$, for the fuzzy threshold as follows:(7)$$e_{k}^{\prime }=\frac{\sum_{i=1}^{n}S_{i}(e_{k})B_{i}(e_{k})}{\sum_{i=1}^{n}S_{i}(e_{k})},$$
where *S_i_* is the membership grade of the *i*th premise in the inference rule, and *B_i_* is the central value of the *i*th conclusion in the inference rule.

The threshold value is then updated by
(8)$$Y(k)=Y(k-1)+e_{k}^{\prime }\times L,$$
(9)$${threshold}_{k}=Y(k)\times 2.$$

Finally, the output is 1 which represents dash, if *I*(*k*) ≥ ${threshold}_{k-1}$; otherwise, the output is 0 which represents dot.

### 2.2. Human–Computer Interface

The human–computer interface layout of AIMcT is shown in Figure 5. The left side of the picture is the mode indicator. The “Record” button will record the operation time data of the practice mode for subsequent system performance analysis and improvement. The right side is the system function settings, including display and operation settings and the “Save” button which can save the settings. When the system is executed again, the previous settings will be automatically loaded. The “hints” field will prompt the user with operation information. For example, the practice mode will prompt the input sequence of long and short sounds of Morse code characters. The keyboard mode will prompt to turn on and off the input mode to prevent accidental input, mode switching codes, etc. The last “output” field can be used as the output display of characters in the practice mode, giving feedback to the user to correct the input method, so as to achieve the learning purpose of becoming familiar with Morse code input.

## 3. Results

### 3.1. AIMcT System

The system designed in this study is shown in Figure 5. It uses a webcam to capture images, the Dlib module to capture facial movement features and detect movement duration, then integrates it with the image Morse code fuzzy recognition system established in this research. Integrate and convert Morse code into corresponding text or instructions: This system has three primary modes: keyboard, mouse, and practice. In addition to being used as a replacement for the keyboard and mouse, the system can also be used to familiarize users with the system through the training mode, and the three modes can be freely switched. This system also includes auxiliary functions such as mouse movement speed adjustment, image size adjustment, automatic image straightening and light compensation, making the system operation more stable and smooth.

#### 3.1.1. Automatic Image Straightening

In the image straightening part, the system will automatically straighten the image based on the confidence value returned from the Dlib function. Figure 6 shows a 45-degree tilted image. The system cannot detect the facial anchor point. After nine adjustments of the system correction function, the system can accurately detect facial anchor points

#### 3.1.2. Automatic Image Compensation

The image automatic compensation capability of the AIMcT system is illustrated in Figure 7. Figure 7a depicts a state with no lighting, where the Dlib module is unable to capture facial features. Following the system’s automatic light compensation calculation, as shown in Figure 7b, the image brightness is enhanced. This enables the Dlib module to function effectively, ensuring the AIMcT system’s proper operation, even during night-time.

### 3.2. AIMcT System Performance Test

In this study, there are twenty-three human-typed data sets: thirteen data sets typed by wireless experts who are skilled in Morse code typing, and the other ten data sets typed by a person with a spinal cord injury. In Figure 8, the average recognition rates of expert data and disabled individuals are 99.83% and 98.6%, respectively. To observe the adjustment threshold of Morse code, we use the disability data set 2 to describe the recognition results, as shown in Figure 9. Figure 9 displays Morse code sequences typed by the disabled person. The symbol definitions are as follows: the ordinate represents the Morse code time length in milliseconds (ms); the abscissa represents the points of Morse code sequences; “Δ” indicates the long element (dash or long-silence); “·” represents the short element (dot or short-silence); “-.” refers to the predictive threshold; the green circle indicates the points of error recognition.

This result shows that through the fuzzy algorithm, disabled people can use this system like experts, and it also proves that the fuzzy algorithm is effective in identifying unstable Morse codes produced by disabled people.

### 3.3. Install and Apply

In order to be able to execute on different operating systems (Arch Linux (×86), Windows 11 (64 bit ×86), and Raspberry Pi OS (64 bit arm64)), this research packages the program code and related packages into program executable files that can be executed on the operating system. When installing in the future, users only need to copy the folder containing the executable file and related resources to the user’s computer. Then, we can use any APPs on PC through operating the AIMcT. Here, Apps is an abbreviation for application. An app is a piece of software. It can run on the Internet, on your computer.

Figure 10 shows the time series and encoded data of a user typing “a” to “z” using AIMcT in keyboard mode through a webcam and output to Microsoft Notepad. In addition, AIMcT also has a mouse function, which can be switched to mouse mode through a mode switching code [29].

## 4. Discussion

Aside from offering an alternative to computer usage for individuals with severe physical disabilities, this study serves the more critical purpose of addressing the limitations of the previously examined MCT system [29].

When comparing MCT and AIMcT in Table 1, the MCT system, built around a single microchip, demands intricate software and hardware design. Its development entails a prolonged period, with a complex production process that presents maintenance challenges. Human resources are needed for both production and upkeep. Additionally, external switch positioning assistance is necessary during use, and system updates are not straightforward.

In contrast, the AIMcT system can be described as application software compatible with both computers and embedded systems. Its development is purely software-based, resulting in significant time savings. Thanks to recent advancements in computer performance, the AIMcT system exhibits remarkable stability and speed, allowing for effortless updates whenever needed. Maintenance is streamlined and hassle-free.

## 5. Conclusions

The AI image Morse code translation system studied in this research allows users to use it without wearing any devices. This breakthrough technology introduces a novel assistive input tool designed to benefit individuals with severe disabilities, granting them unrestricted access to computers.

While Morse code forms the cornerstone of the system, its implementation is greatly facilitated by efficient recognition algorithms equipped with fuzzy time-tracking capabilities. This remarkable feature minimizes the learning curve for users, as their proficiency naturally influences system performance. As the adage goes, practice yields perfection. With dedicated practice, every user can harness this system effectively to communicate and interact with others and enhance their quality of life.

Additionally, this system addresses the longstanding challenge of relying on external assistance for the installation and adjustment of MCT. Moreover, streamlining the production process, reducing costs, simplifying maintenance, and facilitating updates all contribute to a substantially improved adoption rate. Through the utilization of this system, those in need can rediscover the true essence of life.

## Figures and Tables

**Figure 1 bioengineering-10-01281-f001:**
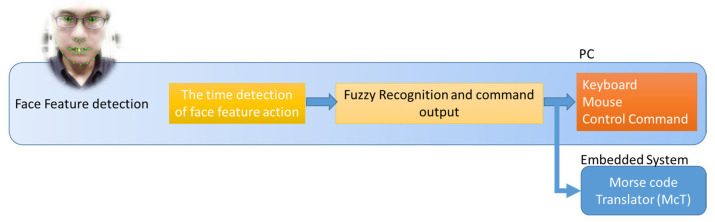
The structure of the AIMcT system.

**Figure 2 bioengineering-10-01281-f002:**
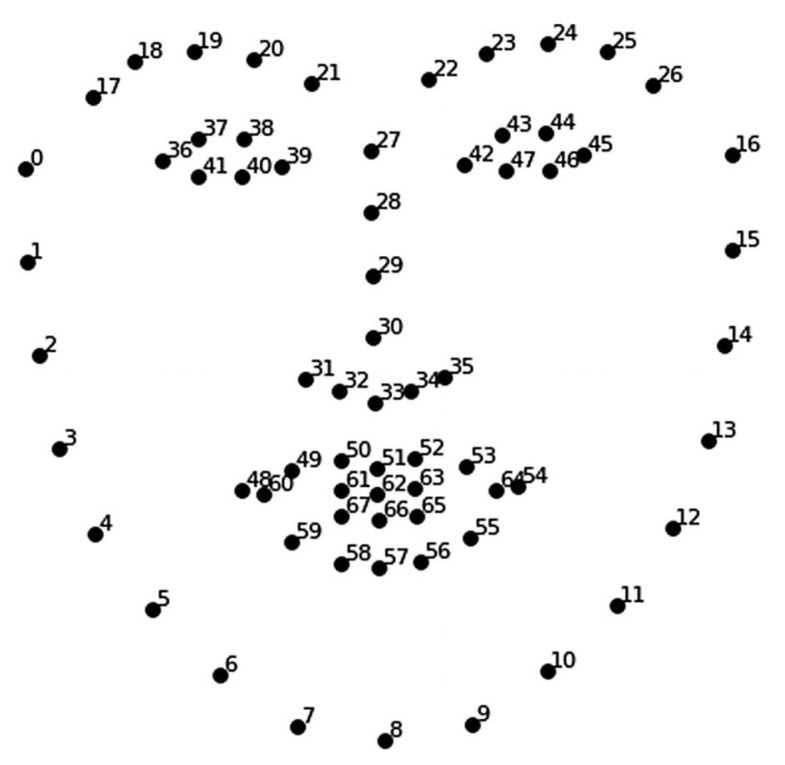
Dlib image recognition module [30].

**Figure 3 bioengineering-10-01281-f003:**
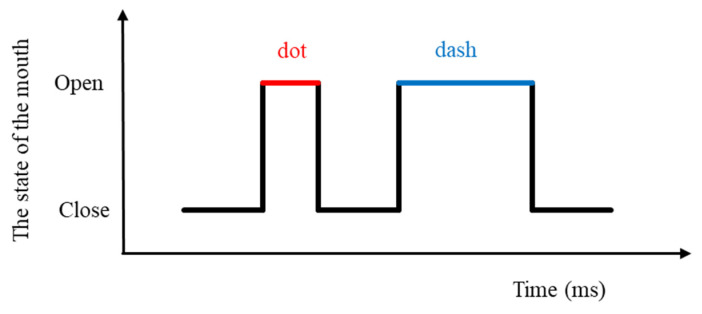
Schematic diagram of continuous mouth movements.

**Figure 4 bioengineering-10-01281-f004:**
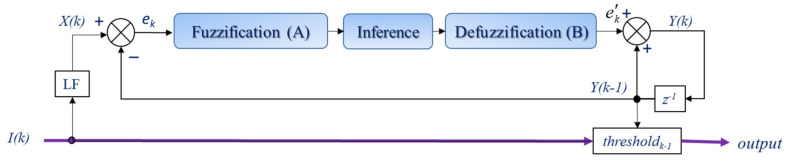
The structure of the FTR.

**Figure 5 bioengineering-10-01281-f005:**
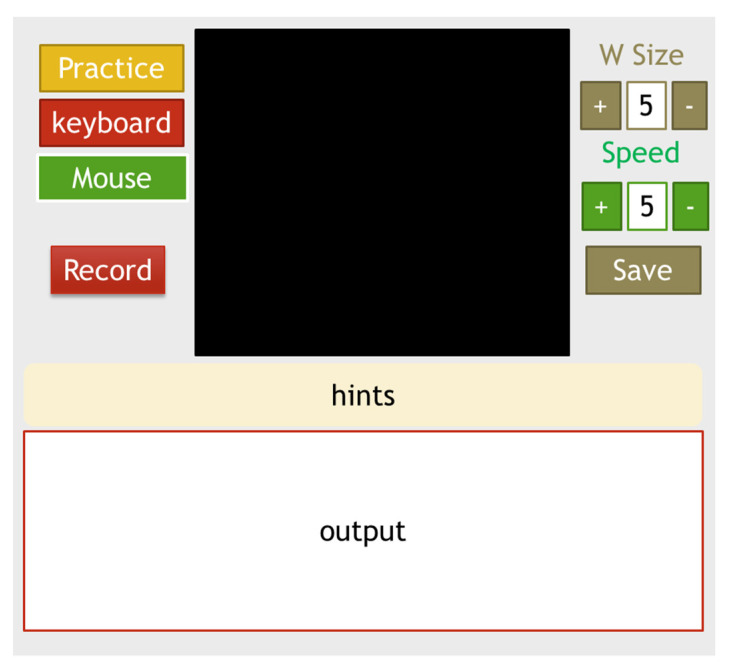
The human–computer interface layout of AIMcT.

**Figure 6 bioengineering-10-01281-f006:**
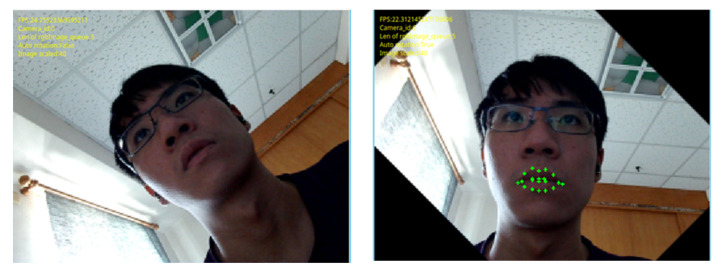
The automatic image straightening performance of the AIMcT system.

**Figure 7 bioengineering-10-01281-f007:**
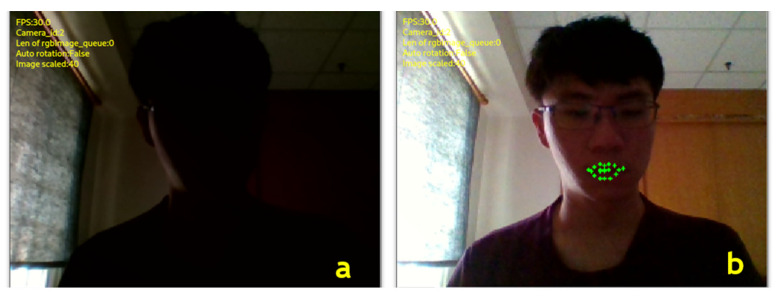
The automatic image compensation performance of the AIMcT system (**a**) before compensation, (**b**) after compensation.

**Figure 8 bioengineering-10-01281-f008:**
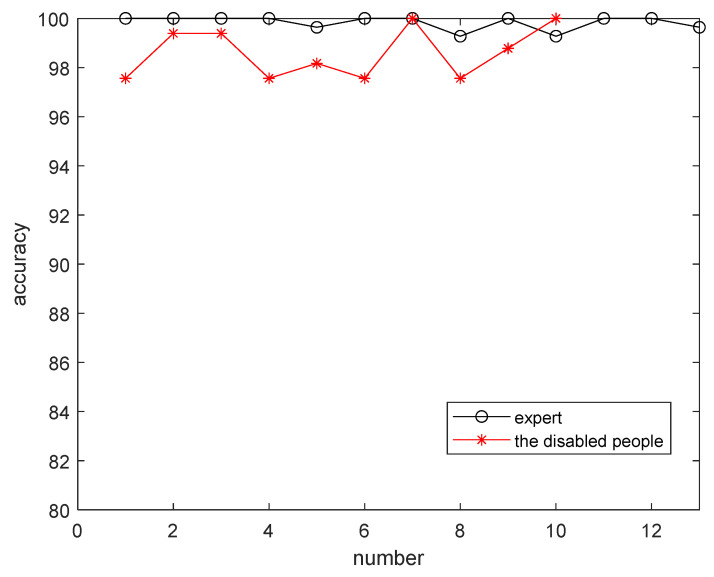
The accuracy of Morse code input by experts and disabled people.

**Figure 9 bioengineering-10-01281-f009:**
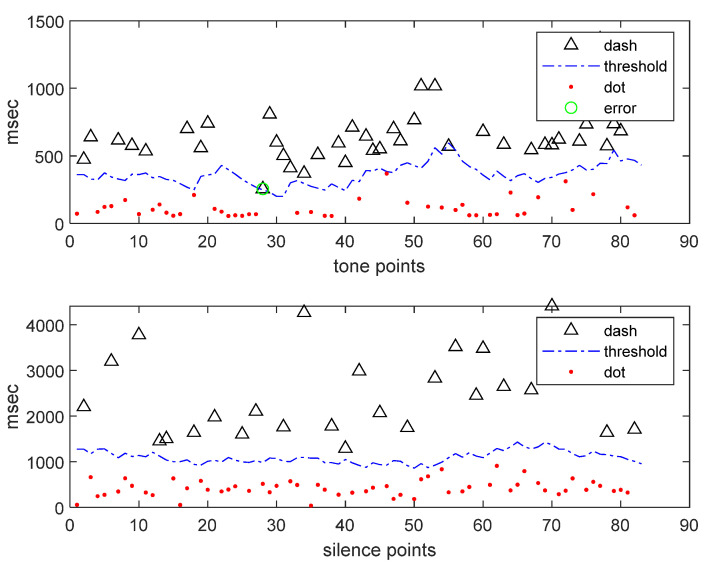
Morse code time series of input “a”–”z”, with adjusted threshold value.

**Figure 10 bioengineering-10-01281-f010:**
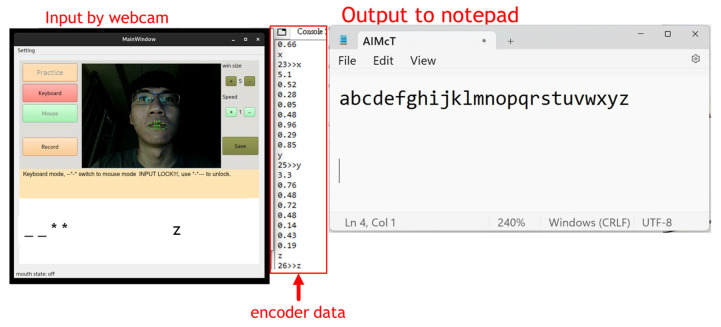
AIMcT keyboard typing mode.

**Table 1 bioengineering-10-01281-t001:** Comparison between MCT and AIMcT.

	MCT	AIMcT
Production	hardware	software
Making process	time consuming	time saving
Price	higher	lower
Core	microprocessor	computer/embedded system
External switch	contact	contactless
Update	difficult	easy
Maintain	difficult	easy

## Data Availability

Not applicable.

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
