# Peer review of "Creating an AI-Enhanced Morse Code Translation System Based on Images for People with Severe Disabilities"

_bioengineering, 2023, doi:10.3390/bioengineering10111281_

Round 1

Reviewer 1 Report

Comments and Suggestions for Authors

This paper presents a set of image-assistive communication equipment based on artificial intelligence to solve communication problems of their daily needs. Using artificial intelligence for facial positioning, and facial motion recognition generates Morse code, and then translates it into readable characters or commands, it allows users to control computer software by themselves and can communicate through wireless networks or Bluetooth protocol to control environment peripherals. The contribution of this paper is prominent. However, there are several flaws in this paper, as follows.

1.     The format in which references are cited in the paper must be carefully revised (line 45).

2.     The feature extraction method of image recognition should be given and improved in this paper.

3.     What is the correlation between image recognition and behavior?

Author Response

To Whom This May Concern,

We must thank the reviewers for their constructive comments! Based on the review results, we were able to significantly improve the manuscript. To help the editor and reviewers go over the revision, the main responses are summarized below.

Reviewer 1:

This paper presents a set of image-assistive communication equipment based on artificial intelligence to solve communication problems of their daily needs. Using artificial intelligence for facial positioning, and facial motion recognition generates Morse code, and then translates it into readable characters or commands, it allows users to control computer software by themselves and can communicate through wireless networks or Bluetooth protocol to control environment peripherals. The contribution of this paper is prominent. However, there are several flaws in this paper, as follows.

  1. The format in which references are cited in the paper must be carefully revised (line 45).

Responses:

   Thank you for your suggestion. I have revised it as suggested.

P.1. “The Policy Department for Economic, Scientific, and Quality of Life Policies of the European Parliament categorized the (CR)s into the following four groups: Robotic surgery, Care and socially assistive robots, Rehabilitation systems, and Training for health and care workers. [28]

  1. The feature extraction method of image recognition should be given and improved in this paper.

Responses:

   Thank you for your suggestion.

   Occasionally, the subject's image placement and the intensity of ambient lighting can impact feature extraction. Within the scope of this study, sections 2.1.1 to 2.1.2 elucidate the enhanced method for feature extraction presented in this research paper.

  1. What is the correlation between image recognition and behavior?

Responses:

Thank you very much.

  P.5. The artificial intelligence-enhanced Morse code translation system's management of image command combinations hinges on the identification of mouth opening and closing states, as well as the duration of the intervals between them. The artificial intelligence Dlib module is used to detect the mouth opening and closing status. To enhance input efficiency and accuracy, and maintain the stability of these command combinations, the system dynamically adjusts the time threshold for opening/closing mouth movements.

Reviewer 2 Report

Comments and Suggestions for Authors

1. in lines 84 and 85, it said "Unfortunately, some of the mentioned AAC devices 84 necessitate users to wear them and rely on a caregiver for their operation. In the absence 85 of assistance, patients are left with no option but to wait." The authors said that some of the devices must be worn, and image recognition research in this area should be given.

2. In Figure 2., the size of 68 feature points is too small.

3. In Figure 3., the font size of the title in this figure is not uniform.

4. in the part of "3.2. AIMcT System Performance Test", the background of "wireless experts" should be given by the authors.

Author Response

To Whom This May Concern,

We must thank the reviewers for their constructive comments! Based on the review results, we were able to significantly improve the manuscript. To help the editor and reviewers go over the revision, the main responses are summarized below.

Reviewer 2:

  1. in lines 84 and 85, it said "Unfortunately, some of the mentioned AAC devices 84 necessitate users to wear them and rely on a caregiver for their operation. In the absence 85 of assistance, patients are left with no option but to wait." The authors said that some of the devices must be worn, and image recognition research in this area should be given.

Responses:

Thank you very much.

In lines 75 to 96, “Eye-tracking technology has shown substantial contributions in various research areas particularly in health care, education, and industrially. Eye tracking has been able to provide valuable support for individuals with severe disabilities by being a useful tool for human-computer interaction. [29] Head operated computer mouse employs two tilt sensors placed in the headset to determine the head position and to function as a simple head-operated computer mouse. One tilt sensor detects the lateral head motion to drive the left/right displacement of the mouse. The other one detects the head's vertical motion to move up and down with respect to the displacement of the mouse. A touch switch device was designed to contact gently with the operator's cheek. The operator may puff his cheek to trigger the device to perform single click, double clicks, and drag commands. [8] An eyeglass-type infrared (IR)-controlled computer interface for the disabled, this system may serve to assist those who suffer from spinal cord injuries or other handicaps in operating a computer. This design use of an infrared remote module fastened to the eyeglasses could allow the convenient control of the input motion on the keys of a computer keyboard and mouse which are all modified with infrared receiving/signal-processing modules. [11] Physiological signals such as EOG, EMG, and EEG can be instrumental in solving communication or computer control problems for severely disabled patients. However, these signals are susceptible to interference from environmental factors, leading to system instability. Scanning auxiliary input tools are polling-based devices, exemplified by the Assistive Context-Aware Toolkit developed by Intel for Dr. Hawking, where users only need to use controllable parts of their body to operate it. One drawback is its relatively slower operation [11].”

In lines 106 to 109, “Regrettably, some of the AAC devices mentioned earlier [5-11, 13-27] necessitate users to wear them and rely on caregivers for operation. In the absence of assistance, patients are left with no alternative but to wait.“

In lines 125 and 126, “The system uses artificial intelligence technology to extract facial features [30, 31]…”

  1. In Figure 2., the size of 68 feature points is too small.

Responses: Thank you very much. I have resized Figure 2 as suggested.

  1. In Figure 3., the font size of the title in this figure is not uniform.

Responses: Thank you very much. I have uniform the font size of the title in Figure 3 as suggested.

  1. in the part of "3.2. AIMcT System Performance Test", the background of "wireless experts" should be given by the authors.

Responses:

Thank you very much.  "wireless experts who are skilled in Morse code typing"

Reviewer 3 Report

Comments and Suggestions for Authors

It is very interesting paper because improved a novel mechanism of comprehensive method of speech that is revolutionary for the disabled people

Author Response

Thank you very much.